# Discriminative Attribute Graph Clustering Through Topology-Guided Contrastive Learning

**Ling Ding** [1]  **Zhizhi Yu** [1 2 *]  **Cuiying Huo** [1 2 *]

## Abstract

Deep attribute graph clustering aims to learn discriminative node representations by leveraging both node attributes and graph topology to partition nodes into distinct clusters. Although substantial progress has been made in attribute-graph clustering in recent years, two key challenges remain: noisy edges in the original adjacency matrix degrade the quality of information propagation, and redundant feature information across different feature views hampers the learning of discriminative representations. To address these issues, we propose a self-supervised attribute graph clustering method based on topological reconstruction and correlation decorrelation. First, we reconstruct the graph topology by computing intersections between k-nearest neighbors and the original adjacency relationships, while simultaneously leveraging global semantic information from K-means clustering to filter out noisy nodes. This reconstructed topology effectively mitigates information redundancy during feature aggregation in Graph Neural Networks. Second, we treat the feature representations from an auto-encoder (AE) and a graph auto-encoder (GAE) as two complementary natural views. We then apply mutual-information-based redundancy reduction and a decorrelation constraint to suppress redundant information between views, yielding more discriminative node representations. Extensive experiments on four widely-used graph datasets—ACM, DBLP, CITE, and AMAP—demonstrate that our method outperforms six representative attributed graph clustering baselines.

[1] College of Intelligence and Computing, Tianjin University, Tianjin, China [2] School of Computer Software, Tianjin University, Tianjin, China. Correspondence to: Zhizhi Yu, Cuiying Huo <yuzhizhi,huocuiying@tju.edu.cn>.

*Proceedings of the $43^{rd}$ International Conference on Machine Learning*, Seoul, South Korea. PMLR 306, 2026. Copyright 2026 by the author(s).

## 1. Introduction

Clustering is a significant unsupervised data mining technique (Ding et al., 2025a;b) that discovers underlying patterns in data without prior knowledge, widely employed in machine learning, image recognition, and computer vision (Jain et al., 1999). With the growth of the Internet and mobile devices, data volume and dimensionality increase substantially, leading to the emergence of deep clustering. Deep clustering leverages deep learning capabilities in feature dimension reduction and representation learning (Xie et al., 2016; Yang et al., 2019), combining deep neural networks with clustering algorithms to enhance performance and effectively handle large-scale, high-dimensional data.

Graph-structured data is pervasive across various domains such as social networks (Jin et al., 2023b), transportation networks (Jin et al., 2025), recommendation systems (Jin et al., 2023a), and molecular structures (Khemani et al., 2024). Graph Neural Networks (GNNs) (Kipf & Welling, 2016) can integrate both graph structure and node attribute information to extract meaningful representations, leading to the exploration of GNN-based deep clustering methods (Ding et al., 2024; 2026b). Deep attribute graph clustering (Lin et al., 2023) is a fundamental and challenging task in machine learning that aims to train neural networks to learn node feature representations from both topological structure and node attributes, ultimately partitioning nodes into different clusters without label information.

Attribute graph clustering has garnered significant attention recently, predominantly through methods based on graph auto-encoders (GAEs) (Bai et al., 2024) and variational graph auto-encoders (VGAEs) (Kipf & Welling, 2016). GAEs generally encode node attributes alongside structural cues via a graph encoder, subsequently reconstructing the topology through an inner-product decoder. Advancing this paradigm, (Guo & Dai, 2022) present a clustering model based on variational graph embeddings, utilizing a GCN-based VGAE (Hamilton et al., 2017; Ding et al., 2026a) to formulate an end-to-end framework enhanced by joint strategies and self-training. Alternatively, (You et al., 2020) develop a graph contrastive learning framework that integrates data augmentation strategies (e.g., node dropping, edge perturbation, attribute masking, and subgraph sam-

pling). By maximizing consistency between dual views via a contrastive objective, their method generates feature representations characterized by stronger generalization capabilities and robustness.

GNNs effectively encode both node attributes and topology, yielding promising results in attribute graph clustering. However, the field still faces several challenges: 1) The given adjacency matrix incorporates neighboring node features regardless of their categories. The challenge is how to fully explore data relationships, reconstruct graph topology, and generate more meaningful clustering representations. 2) How to overcome issues in data-augmentation-based contrastive learning methods, reduce model complexity, and eliminate redundant features while fully exploiting the data.

To address these challenges, we propose a self-supervised attribute graph clustering method based on topological reconstruction. We compute node similarity using the feature matrix, determine k nearest neighbors, and filter out noisy nodes by leveraging local structural and global semantic information to reconstruct adjacency relationships, benefiting discriminative node feature learning. Unlike previous data augmentation approaches, we utilize embeddings from an Auto-Encoder (AE) and a graph auto-encoder as two views for self-supervised contrastive learning. By incorporating mutual-information-based redundancy reduction constraints and decorrelation operations, we extract rich information while eliminating redundancy, resulting in more discriminative node representations.

The main contributions of this paper are as follows:

- We propose a topology reconstruction method that integrates local structural and global semantic information to generate more discriminative node representations, thereby improving graph clustering.
- We develop a contrastive learning approach that does not require data augmentation, utilizing mutual-information-based redundancy reduction and decorrelation constraints to reduce redundancy and enhance performance.
- We conduct extensive experiments on four popular graph datasets, including ACM, DBLP, CITE, and AMAP, demonstrating the effectiveness of the proposed method.

## 2. Related Work

**Attributed Graph Clustering**. An attributed graph can be represented as $\mathcal{G} = (\mathcal{V}, \mathcal{E}, \mathcal{X})$, where $\mathcal{V} = \{v_1, v_2, \ldots, v_n\}$ represents the set of $n$ nodes, $\mathcal{E}$ represents the set of edges, and $\mathcal{X}$ represents the node attributes. The node feature matrix is denoted as $X \in \mathbb{R}^{n \times d}$, where $d$ is the feature dimension. The topological structure is represented by the adjacency matrix $\mathbf{A} \in \mathbb{R}^{n \times n}$, where $A_{i,j} = 1$ if edge $e_{i,j} \in \mathcal{E}$, and $A_{i,j} = 0$ otherwise.

Attribute graph clustering encodes node attribute information using graph topological structure to partition nodes into disjoint clusters (Wu et al., 2021). (Wang et al., 2019) introduce a unified framework, DAEGC, that jointly optimizes feature extraction and clustering modules through a graph auto-encoder with attention mechanisms. To address GNNs' over-smoothing issues that diminish discriminative node representations, SDCN (Bo et al., 2020) and (Li et al., 2018) combine auto-encoders and graph auto-encoders for effective feature learning. AGCN (Peng et al., 2021) employs two attention-based fusion modules for heterogeneity and multi-scale feature fusion to learn information-rich representations. DAGC (Peng et al., 2023) builds upon AGCN by incorporating a distribution fusion module and attention-based mechanisms, improving clustering through soft self-supervision and hard self-supervision. DFCN (Tu et al., 2021) extends SDCN with a fusion module that combines structural and attribute information, thereby promoting better AE-GAE information exchange and improving clustering outcomes.

**Self-supervised Learning**. Due to limitations of supervised and semi-supervised learning methods, such as poor generalization and label dependence, self-supervised learning has gained significant attention. It relies on well-designed proxy tasks without manual annotations, utilizing inherent data information to acquire knowledge. Self-supervised learning can be categorized into four types: generation-based, auxiliary property-based, contrast-based, and hybrid methods (Liu et al., 2023). Among these, contrast-based methods are more flexible and widely employed. A typical graph contrast model consists of three sequential modules: view augmentation, view encoding, and representation contrasting (Xu et al., 2021).

Deep graph infomax (DGI) (Veličković et al., 2018) applies mutual information maximization to graph data, learning node feature representations by comparing local node-level with global graph-level representations. Building upon DGI, GMI (Peng et al., 2020) directly maximizes mutual information between input and output of the graph encoder, considering both node features and topological structure. (Zhu et al., 2021) introduced an adaptive augmentation graph contrastive learning method that removes unimportant edges and perturbs unimportant feature dimensions to generate two different graph views. A contrastive learning objective then learns consistency of each node's feature representation across these views using a mutual information estimator based on noise-contrastive estimation.

$$\mathcal{L}_{NCE}\left(p_\varphi\left(h_i, h_j\right)\right) = -\mathcal{MI}_{NCE}\left(h_i, h_j\right)$$
$$= -\mathbb{E}_{\mathcal{P} \times \tilde{\mathcal{P}}^N}\left[\log \frac{e^{p_\varphi(h_i, h_j)}}{e^{p_\varphi(h_i, h_j)} + \sum_{n \in N} e^{p_\varphi(h_i, h'_n)}}\right], \quad (1)$$

where $(h_i, h_j)$ represents the feature representation obtained through feature learning for a given instance $(x_i, x_j)$, $p_\varphi(h_i, h_j)$ is defined as $h_i^T h_j / \tau$, where $\tau$ represents the

temperature parameter.

Barlow Twins ([Bielak et al., 2022](#)) proposes a self-supervised framework that does not require negative samples. Given two augmented views of the same batch, the model learns invariant and non-redundant representations by forcing the cross-correlation matrix between the two views to be close to the identity matrix. Specifically, the diagonal terms of the cross-correlation matrix are encouraged to approach one, improving the consistency between the two views, while the off-diagonal terms are penalized to reduce redundancy among different feature dimensions.

$$\mathcal{L}_{BT}\left(H^{(1)}, H^{(2)}\right)$$
$$= \mathbb{E}_{\mathcal{B} \sim \mathcal{P}^{|\mathcal{B}|}} \left[ \sum_a \left(1 - C_{aa}\right)^2 + \lambda \sum_a \sum_{b \neq a} C_{ab}^2 \right], \quad (2)$$

where $a$ and $b$ represent the feature dimensions while $i$ represent the samples in batch $\mathcal{B}$, the cross-correlation matrix is defined as $C_{ab} = \frac{\sum_{i \in \mathcal{B}} \hat{H}_{ia}^{(1)} \hat{H}_{ib}^{(2)}}{|\mathcal{B}|}$, with $\hat{H}_{ia}^{(1)} = \frac{H_{ia}^{(1)} - \mu_a^{(1)}}{\sigma_a^{(1)}}$ and $\hat{H}_{ib}^{(2)} = \frac{H_{ib}^{(2)} - \mu_b^{(2)}}{\sigma_b^{(2)}}$, and $\mu_a^{(1)}$ and $\sigma_a^{(1)}$ are the mean and standard deviation of the $a$-th feature dimension in batch $\mathcal{B}$.

## 3. The Proposed Method

In this section, we propose a self-supervised attribute graph clustering model based on topological reconstruction to address the aforementioned issues. This model effectively integrates the local structural information and the global semantic information of the graph by reconstructing its topological structure. Through contrastive learning, the model learns more discriminative feature representations, thereby improving clustering performance. The overall network architecture, as shown in Figure 1, consists of four main modules. The graph auto-encoder module captures the structural information of the data by reconstructing the graph. The auto-encoder module extracts the content information of the data. The contrastive learning module compares the feature representations learned from the two aforementioned modules to obtain more distinguishable node features. Finally, the self-supervised clustering module is specifically designed for the clustering task.

### 3.1. Topological Reconstruction

The information propagation process in GNNs fundamentally involves the fusion of features from neighboring nodes. However, when two nodes connected by an edge do not belong to the same class, the feature fusion process can introduce redundant information, which affects the clustering accuracy. Hence, it becomes essential to reconstruct a graph structure that is conducive to learning discriminative features. Augmentation-free graph self-supervised learning

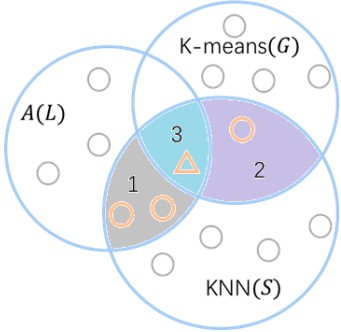

*Figure 1.* Illustration of adjacent node selection.

methods ([Huang et al., 2018](#); [Lee et al., 2022](#)) exploit local and global information to construct positive pairs between online and target representations. Inspired by this, we utilize the local structural information and global semantic information of the samples to construct a graph structure for feature learning. The method of constructing the graph structure involves defining the relationships between samples to determine whether there is an edge connecting them. First, we calculate the cosine similarity between every pair of nodes utilizing the feature matrix $X$. This computation can be performed as follows:

$$Sim = \frac{X X^T}{||X|| \cdot ||X||}, \quad (3)$$

Based on the similarity matrix $Sim$, we search for k nearest neighbors for each node $v_i$, and the result is represented by the set $S_i$, as shown in the region $KNN(S)$ in Figure 2. The aforementioned calculation only ensures that the nodes in the set $S_i$ are similar to $v_i$ at the feature level. However, the set $S_i$, as a reasonable set of adjacent nodes to $v_i$, may contain inherent noisy nodes. To overcome the limitations of relying solely on finding nearest neighbors in the sample space, which disregards the inherent structural information in the graph and may include semantically unrelated samples to the query node $v_i$, the following strategies are employed. These strategies aim to capture both local structural information and global semantic information of the graph while filtering out false adjacent nodes in the process.

First, we capture local structural information between nodes provided by the adjacency matrix $A$. Based on the smoothness assumption ([Zhu et al., 2003](#)), for a node $v_i$, its set of neighboring nodes $L_i$ tends to have the same label as $v_i$, as shown in the $A(L)$ region in Figure 2. The nodes in $L_i$ are the ones connected to the query node $v_i$ in the adjacency matrix $A$. Therefore, to capture local structural information and filter out noisy nodes in the set $S_i$, we compute the intersection between the k-nearest neighbors and the set of neighboring nodes, $S_i \cap L_i$, as depicted in the regions labeled 1 and 3 in Figure 2.

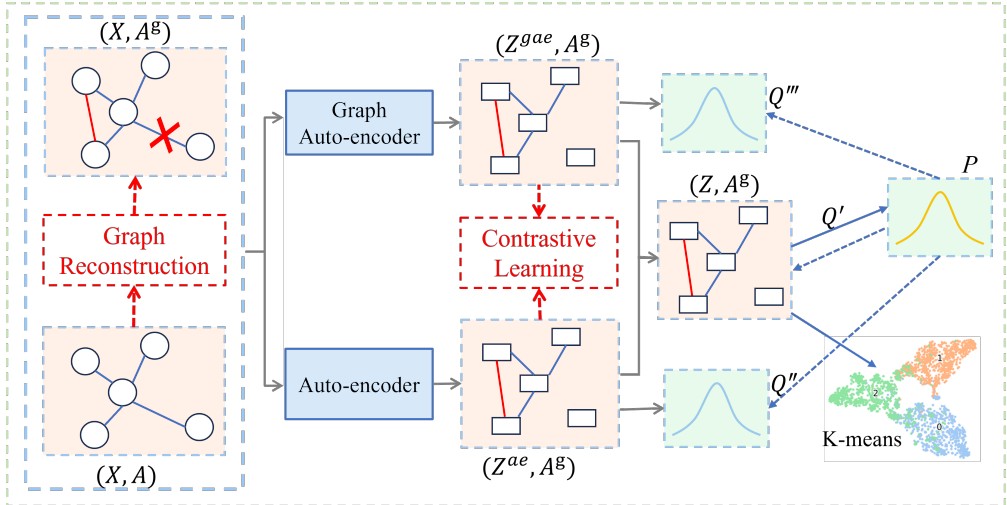

*Figure 2.* An overview of self-supervised attribute graph clustering based on topological reconstruction.

Next, we focus on capturing the global semantic information. To achieve this, we employ clustering methods that help capture the semantic information of nodes at a global level. The goal is to identify non-adjacent nodes that exhibit similar global semantic characteristics as the query node $v_i$. Although some nodes may not be connected by edges, they may have the same label information. We consider these nodes to be semantically similar but not directly connected, and we can discover them through clustering methods on a global scale. We apply the K-means algorithm (Hartigan & Wong, 2018) on the feature matrix X to cluster nodes into $K$ clusters. Then, we compute the set $G_i$ of nodes that belong to the same cluster as $v_i$, and we consider $G_i$ as the set of nodes semantically similar to $v_i$ from a global perspective, as shown in the K-means($G$) region in Figure 1. Therefore, to capture the global semantic information and filter out noisy nodes in the set $S_i$, we compute the intersection between the k-nearest neighbors and the set of globally semantically similar nodes, denoted as $S_i \cap G_i$, as depicted in the regions labeled 2 and 3 in Figure 2.

To simultaneously consider the local structural information and global semantic information of the graph, we define the set of adjacent nodes as $D_i = (S_i \cap L_i) \cup (S_i \cap G_i)$, as shown in the regions labeled 1, 2, and 3 in Figure 2. We then construct the topological structure of the graph based on the set $D$ for subsequent feature learning.

### 3.2. Graph Auto-encoder Module

GCN is capable of effectively processing both the structural and content information of the data, leading to improved clustering performance. In light of this, we employ a graph auto-encoder based on GCN to reconstruct the features and structure of the data.

The convolution operation of the graph auto-encoder can be described as follows:

$$Z^{(l)} = \emptyset(\tilde{D}^{-\frac{1}{2}} \tilde{A} \tilde{D}^{-\frac{1}{2}} Z^{(l-1)} W^{(l-1)}), \qquad (4)$$

where $\tilde{A} = A^g + I$, $\tilde{D}_{ii} = \sum_j \tilde{A}_{ij}$ and $I$ be the identity matrix. The matrix $A^g$ represents the reconstructed adjacency matrix obtained in Section 3.1, while $\tilde{A}$ denotes the adjacency matrix with a self-loop. The activation function is represented by $\emptyset$.

The reconstruction loss in this part is given by:

$$\mathcal{L}_{GAE} = \mathcal{L}_{GAE^g} + \mathcal{L}_{GAE^f}, \qquad (5)$$

The first term of Eq.5 is denoted as $\mathcal{L}_{GAE^g} = ||A^g - A'||_F^2$, where $A' = Sigmoid((Z^{gae})^T Z^{gae})$ and $Z^{gae}$ represents the output of the last layer of the graph encoder. The feature reconstruction loss represented by $L_{GAE^f} = ||X - X'||_F^2$, where $X'$ is the reconstructed feature matrix obtained from the graph auto-encoder.

### 3.3. Auto-encoder Module

Deep convolutional neural networks (CNNs) are highly effective in extracting informative features from complex data. In our method, we utilize an auto-encoder built upon a deep CNN to reconstruct node features. By leveraging the power of deep CNNs, the auto-encoder can capture and represent essential characteristics of the data, leading to improved clustering results.

The information propagation process of the auto-encoder is:

$$H^{(l)} = a_l(w_l H^{(l-1)} + b_l), \qquad (6)$$

where $w_l$ is the weight of the $l$-th layer, $b_l$ is the bias of the $l$-th layer, $a_l$ is the activation function of the $l$-th layer, $H^{(l)}$ is the embedding representation of the $l$-th layer.

This part of the reconstruction loss is the feature reconstruction loss, as follows:

$$\mathcal{L}_{AE} = ||X - \hat{X}||_F^2, \tag{7}$$

where $\hat{X}$ represents the reconstructed feature matrix of the auto-encoder.

### 3.4. Contrastive Learning Module

In order to filter out redundant information, we adopt a self-supervised contrastive learning method to learn discriminative features. However, most existing methods, such as (Bielak et al., 2022; Liu et al., 2022; He et al., 2023), often employ data augmentation to contrast the two learned feature representations. This approach introduces additional temporal and spatial complexity to the model. Additionally, the performance of the model is greatly influenced by the selection of the augmentation scheme. Without carefully designed augmentation techniques, arbitrary graph expansion can result in substantial alterations to the underlying semantics of the graph. To tackle these challenges, we consider the feature representations obtained from GAE in Section 3.2 and the feature representations acquired from AE in Section 3.3 as two different views of the original data. We introduce the constraint of minimizing mutual information between these two views and employ a decorrelation loss function to accomplish this objective.

For the purpose of minimizing the mutual information between the two views, we employ the method of Mutual Information Neural Estimation (MINE) (Belghazi et al., 2018). MINE trains a classifier to differentiate samples from the joint distribution and the product of marginals of the two random variables. MINE employs an exact estimation formula for mutual information based on the KL divergence. However, since our goal is to minimize mutual information rather than obtain its precise value, we utilize an estimator based on the Jensen-Shannon (JS) divergence (Nowozin et al., 2016).

$$I(Z^{gae}; Z^{ae}) = -sp(-D_\varpi(Z^{gae}, Z^{ae})) \\ - \mathbb{E}_p[sp(D_\varpi(Z^{gae}, \tilde{Z}^{ae}))], \tag{8}$$

where $Z^{gae}$ and $Z^{ae}$ denote the representations learned by the GAE and AE encoders, respectively. $\tilde{Z}^{ae}$ denotes a negative sample drawn from the marginal distribution of the AE-view representations, which is used to approximate the product of marginals.

For the decorrelation constraint, we first calculate the cosine similarity between samples from the two views to obtain the correlation matrix $B$:

$$B_{ij} = \frac{(Z_i^{ae})(Z_j^{gae})^T}{||Z_i^{ae}|| \cdot ||Z_j^{gae}||}, \forall i, j \in [1, N], \tag{9}$$

Then we aim to fit the correlation matrix $B$ to the identity matrix:

$$\mathcal{L}_r = \frac{1}{N^2} \sum (B_{ij} - I)^2 \\ = \frac{1}{N} \sum_{i=1}^N (B_{ii} - 1)^2 + \frac{1}{N^2 - N} \sum_{i=1}^N \sum_{j \neq i} (B_{ij})^2, \tag{10}$$

where $I$ is an identity matrix with the same shape as matrix $B$. The first term encourages the diagonal elements of matrix $B$ to be equal to 1, which promotes the consistency of each node's feature representation between two views. The second term minimizes the off-diagonal elements of matrix $B$ to be equal to 0, aiming to minimize the consistency of different nodes' feature representations between two views. This decorrelation operation helps optimize the network by reducing redundant information in the latent feature space, resulting in more discriminative learned feature representations. The loss function can be defined as follows:

$$\mathcal{L}_{con} = \mathcal{L}_r - I(Z^{gae}; Z^{ae}), \tag{11}$$

### 3.5. Self-Supervised Clustering Module

The above module can extract informative and discriminative features, but the learning process is not directly guided by the clustering task. In order to learn more suitable feature representations for clustering, we introduce clustering constraints into the model optimization process.

First, we calculate the similarity between the i-th sample and the j-th cluster using the Student's t-distribution, obtaining the probability distribution $Q' = [q_{ij}]_{mxn}$:

$$q_{ij} = \frac{(1 + ||z_i - u_j||^2/\alpha)^{-\frac{\alpha+1}{2}}}{\sum_{j'} (1 + ||z_i - u_{j'}||^2/\alpha)^{-\frac{\alpha+1}{2}}}, \tag{12}$$

where $z_i$ is a more robust feature representation obtained by combining $z_i^{ae}$ and $z_i^{gae}$. $u_j$ represents the centroid vector of the $j$-th cluster, and $\alpha$ is the degrees of freedom for the t-distribution, set to 1. Furthermore, according to Eq.12, we compute three soft assignment distributions $Q'$, $Q''$, and $Q'''$ based on the fused representation $z_i$, the AE representation $z_i^{ae}$, and the GAE representation $z_i^{gae}$, respectively.

Next, by sharpening the distribution $Q'$ using Eq.13, we obtain a target distribution $P$ with higher confidence.

$$p_{ij} = \frac{q_{ij}^2 / \sum_i q_{ij}}{\sum_{j'} q_{ij'}^2 / \sum_i q_{ij'}}, \tag{13}$$

Finally, by minimizing the KL divergence between the two distributions using Eq.14, we ensure that the samples in the embedding space are closer to the cluster centers. In this case, $Q = (Q' + Q'' + Q''')/3$.

$$\mathcal{L}_{clu} = KL(P||Q) = \sum_i \sum_j p_{ij} \log \frac{p_{ij}}{\hat{q}_{ij}}, \tag{14}$$

where $Q_{ij}$ denotes the $(i,j)$-th element of the fused soft assignment distribution $Q$.

### 3.6. Objective Function

The overall objective function of the proposed method consists of four components: the reconstruction loss of AE, the reconstruction loss of GAE, the contrastive loss, and the clustering loss.

$$\mathcal{L} = \mathcal{L}_{AE} + \mathcal{L}_{GAE} + \mathcal{L}_{con} + \mathcal{L}_{clu}. \qquad (15)$$

Algorithm 1 provides a detailed description of the learning process of the proposed algorithm.

---

**Algorithm 1** Self-Supervised Attribute Graph Clustering Based on Topological Reconstruction

---

**Require:** Feature matrix $X$, adjacency matrix $A$, number of clusters $N$, number of iterations $E$.
**Ensure:** Cluster result $R$.
 1: Pretrained baseline model obtained feature representation $Z$.
 2: Perform K-means clustering on $Z$, initializing cluster centroids.
 3: Reconstruct the graph structure $A^g$ using the feature matrix $X$ and adjacency matrix $A$.
 4: **for** $i = 1$ to $E$ **do**
 5:   Encode the data using the auto-encoder and graph auto-encoder to obtain the encoded representations $Z^{ae}$ and $Z^{gae}$, respectively.
 6:   Calculate the losses $\mathcal{L}_{AE}$, $\mathcal{L}_{GAE}$, $\mathcal{L}_{con}$, and $\mathcal{L}_{clu}$, respectively.
 7:   Update the network parameters by minimizing the objective function.
 8: **end for**
 9: Perform K-means clustering on the learned feature representations $Z$ to obtain the cluster result $R$.
10: Return $R$.

---

| Dataset | Samples | Dimensions | Edges | Classes |
|---------|---------|------------|-------|---------|
| DBLP | 4057 | 334 | 3528 | 4 |
| CITE | 3327 | 3703 | 4552 | 6 |
| ACM | 3025 | 1870 | 13128 | 3 |
| AMAP | 7650 | 745 | 119081 | 8 |

*Table 1.* Dataset statistics.

## 4. Experiments

We first describe the baselines, dataset statistics, parameter configurations, and cluster evaluation metrics. Next, we compare the clustering performance of our approach against six state-of-the-art baseline methods on four widely-used graph datasets. Additionally, we perform rigorous ablation studies to validate the necessity of each proposed component

and provide t-SNE visualizations to qualitatively demonstrate the effectiveness of our method.

### 4.1. Baselines

To verify the effectiveness of the proposed method, this section performed enriched experiments and compared them with six benchmark methods, including GAE (Kipf & Welling, 2016), DAEGC (Wang et al., 2019), MVGRL (Hassani & Ahmadi, 2020), SDCN (Bo et al., 2020), AGCN (Peng et al., 2021), and DFCN (Tu et al., 2021).

Among them, GAE uses the graph auto-encoder to encode the attribute information and the topology structure of nodes, and uses the inner product decoder to decode the graph structure. DAEGC uses a graph auto-encoder based on an attention mechanism for feature extraction and performs model optimization by constructing a unified framework for joint reconstruction loss and cluster loss. MVGRL treats the node adjacency matrix and the graph diffusion matrix as two graph structure views, by maximizing the mutual information of the node representation of one view and the graph-level representation of the other view. SDCN uses the auto-encoder and GCN to extract both attribute information and structural information. AGCN designs two attention-based fusion modules while considering heterogeneity and multi-scale feature fusion to learn information-rich feature representations adaptively. The DFCN designs a dynamic cross-modal fusion mechanism that generates a more robust target distribution.

The experiment is implemented on Windows 11, Intel Core i7-12700H CPU, NVIDIA GeForce RTX 3060 GPU, based on Python and the PyTorch framework.

### 4.2. Datasets

In this paper, comprehensive experiments are performed to evaluate the effectiveness of the proposed method using four widely used graph datasets: ACM (Bo et al., 2020), DBLP (Bo et al., 2020), CITE (Bo et al., 2020), and AMAP (Liu et al., 2022). The specific information about each dataset is shown in Table 1.

### 4.3. Parameters Setting and Cluster Evaluation Metrics

For the baseline methods GAE (Kipf & Welling, 2016), DAEGC (Wang et al., 2019), MVGRL (Hassani & Ahmadi, 2020), SDCN (Bo et al., 2020), and DFCN (Tu et al., 2021), we refer to the results reported in DCRN (Liu et al., 2022). For the baseline method AGCN (Peng et al., 2021), we refer to the results reported in DAGC (Peng et al., 2023). As for our proposed method, we use DFCN as the backbone network. In all experiments, we train the network using the Adam optimizer (Kingma & Ba, 2015). The experimental settings for the DBLP, CITE, ACM, and AMAP datasets are shown in Table 3.

| Dataset | metric | GAE | DAEGC | MVGRL | SDCN | AGCN | DFCN | OURS |
|---------|--------|-----|-------|-------|------|------|------|------|
| DBLP | ACC | 61.21±1.22 | 62.05±0.48 | 42.73±1.02 | 68.05±1.81 | 73.26±0.37 | 76.00±0.80 | **80.89±0.42** |
| | NMI | 30.80±0.91 | 32.49±0.45 | 15.41±0.63 | 39.50±1.34 | 39.68±0.42 | 43.70±1.00 | **50.78±0.60** |
| | ARI | 22.02±1.40 | 21.03±0.52 | 8.22±0.50 | 39.15±2.01 | 42.49±0.31 | 47.00±1.50 | **56.25±0.77** |
| | F1 | 61.41±2.23 | 61.75±0.67 | 40.52±1.51 | 67.71±1.51 | 72.80±0.56 | 75.70±0.80 | **80.43±0.42** |
| CITE | ACC | 61.35±0.80 | 64.54±1.39 | 68.60±0.36 | 65.96±0.31 | 68.79±0.23 | 69.50±0.20 | **69.98±0.22** |
| | NMI | 34.63±0.65 | 36.41±0.86 | 43.66±0.40 | 38.71±0.32 | 41.54±0.30 | **43.90±0.20** | 43.44±0.31 |
| | ARI | 33.55±1.18 | 37.78±1.24 | 44.27±0.73 | 40.17±0.43 | 43.79±0.31 | 45.50±0.30 | **45.55±0.37** |
| | F1 | 57.36±0.82 | 62.20±1.32 | 63.71±0.39 | 63.62±0.24 | 62.37±0.21 | 64.30±0.20 | **65.14±0.26** |
| ACM | ACC | 84.52±1.44 | 86.94±2.83 | 86.73±0.76 | 90.45±0.18 | 90.59±0.15 | 90.90±0.20 | **91.94±0.24** |
| | NMI | 55.38±1.92 | 56.18±4.15 | 60.87±1.40 | 68.31±0.25 | 68.38±0.45 | 69.40±0.40 | **71.77±0.55** |
| | ARI | 59.46±3.10 | 59.35±3.89 | 65.07±1.76 | 73.91±0.40 | 74.20±0.38 | 74.90±0.40 | **77.59±0.59** |
| | F1 | 84.65±1.33 | 87.07±2.79 | 86.85±0.72 | 90.42±0.19 | 90.58±0.17 | 90.80±0.20 | **91.90±0.26** |
| AMAP | ACC | 71.57±2.48 | 76.44±0.01 | 45.19±1.79 | 53.44±0.81 | 58.53±1.74 | 76.88±0.80 | **79.31±0.59** |
| | NMI | 62.13±2.79 | 65.57±0.03 | 36.89±1.31 | 44.85±0.83 | 51.76±3.23 | 69.21±1.00 | **73.42±0.60** |
| | ARI | 48.82±4.57 | 59.39±0.02 | 18.79±0.47 | 31.21±1.23 | 41.15±2.78 | 58.98±0.84 | **62.73±0.62** |
| | F1 | 68.08±1.76 | 69.97±0.02 | 39.65±2.39 | 50.66±1.49 | 43.68±5.08 | 71.58±0.31 | **73.34±0.35** |

*Table 2.* Clustering results on four datasets.

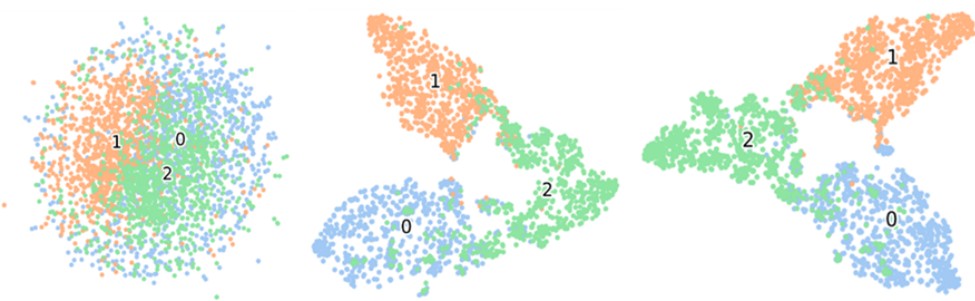

*Figure 3.* The t-SNE visualization on the ACM dataset.

| Dataset | LR | $k$ | $K$ | $neg$ |
|---------|-----|-----|-----|-------|
| DBLP | 1e-4 | 1 | 90 | 5 |
| CITE | 1e-4 | 5 | 200 | 5 |
| ACM | 1e-5 | 1 | 100 | 6 |
| AMAP | 1e-3 | 2 | 100 | 6 |

*Table 3.* Parameter setting. LR is short for Learning Rate.

Among them, k represents the number of neighbors mentioned in section 3.1, K represents the number of cluster centers in the K-means method, and neg represents the number of negative samples selected when calculating mutual information. To avoid the randomness of the clustering results, the reported results are the averages and corresponding standard deviations over 10 runs.

In order to compare the superiority of algorithms, four commonly used clustering metrics (van der Maaten & Hinton, 2008) are used to evaluate the clustering performance: Accuracy (ACC), Normalized Mutual Information (NMI), Ad-

justed Rand Index (ARI), and Macro F1 Score (F1).

ACC is the ratio of assigning the correct number of labels and the number of samples, used to measure the accuracy of the clustering results. NMI represents the amount of correct clustering information included in the clustering results. ARI measures the similarity between two clusters to evaluate the clustering results. F1 refers to the evaluation metric obtained by the accuracy and recall of the weighted clustering results. Among these four metrics, larger values correspond to better clustering results.

### 4.4. Analysis of the Cluster Results

Table 2 reports the clustering performance of all comparative methods on the four commonly used graph datasets. Bold font indicates the best results. From Table 2, it can be observed that the proposed method achieves the best performance on most evaluation metrics across the four datasets. Specifically, on the DBLP dataset, the proposed method

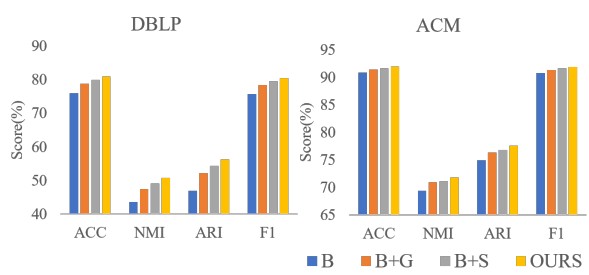

*Figure 4.* Ablation study on DBLP and ACM datasets.

| Method | GAE | DAEGC | MVGRL | SDCN | AGCN | DFCN |
|--------|-----|-------|-------|------|------|------|
| Ours | 0.001 | 0.001 | 0.001 | 0.001 | 0.008 | >0.05 |

*Table 4.* Significance test results.

achieves improvements of 4.89%, 7.08%, 9.25%, and 4.73% in terms of ACC, NMI, ARI, and F1, respectively, compared to DFCN.

To visually illustrate the effectiveness of the proposed method, we utilize the t-SNE algorithm (Demsar, 2006) to visualize the clustering results of the ACM dataset in a two-dimensional space. The visualization includes the original data, the feature representations obtained from the baseline algorithm DFCN, and the feature representations acquired from the proposed method. From Figure 3, it is evident that the positions of the original data points exhibit overlapping, whereas the proposed method successfully separates nodes from different classes and produces more compact intra-cluster structures.

### 4.5. Ablation Study

To rigorously validate the effectiveness and contribution of each proposed component, we conduct comprehensive ablation experiments on the DBLP dataset (Figure 4). We evaluate four configurations: (B) the baseline DFCN model, (B+G) baseline with topological reconstruction module, (B+S) baseline with contrastive learning module, and (OURS) the full model integrating both modules.

The results demonstrate that both components contribute meaningfully to performance improvement. Specifically, adding the topological reconstruction module (B+G) yields improvements across different metrics, indicating that reconstructing the graph topology through local structural and global semantic information effectively mitigates the impact of noisy edges and improves the quality of graph convolution operations. Incorporating the contrastive learning module (B+S) further improves clustering performance, suggesting that the redundancy-reduction regularizer and decorrelation loss help suppress redundant information between the AE and GAE views.

Most importantly, the full model (OURS) achieves the largest performance (compared to DFCN baseline), which is not merely the sum of the individual module contributions. This synergistic effect reveals that topological reconstruction and contrastive learning are complementary: the re-

constructed topology provides cleaner structural signals for the graph auto-encoder, while the decorrelated embeddings from contrastive learning better leverage the refined graph structure during information propagation.

### 4.6. Significance Test

To compare the proposed method with GAE, DAEGC, MV-GRL, SDCN, AGCN, and DFCN, we selected the values of ACC, NMI, ARI, and F1 from Table 2 for four datasets (DBLP, CITE, ACM, and AMAP). We conducted Friedman and Post-hoc Nemenyi tests (Pupo et al., 2018), and the results are presented in Table 4. If the p-value of the test result is less than 0.05, it indicates a significant difference in the experimental results between the two methods. The results presented in Table 4 highlight significant differences between the proposed method and existing approaches such as GAE, DAEGC, MVGRL, SDCN, and AGCN. Although there is no significant difference compared to the baseline method DFCN, the proposed method outperforms DFCN in 15 out of 16 experimental results, except for NMI on the CITE dataset. Although the improvement over DFCN is not statistically significant under the Nemenyi test, our method achieves better results on 15 out of 16 dataset-metric combinations, indicating its competitive and generally stronger empirical performance.

## 5. Conclusion

This paper proposes a self-supervised attribute graph clustering model based on topological reconstruction. The model first integrates local structural information and global semantic information to reconstruct a graph structure suitable for learning discriminative features. Then, it employs two embedding features learned by an auto-encoder and a graph auto-encoder as two views for contrastive learning. By incorporating mutual-information-based redundancy reduction constraints and decorrelation loss constraints, redundant information can be effectively eliminated, and discriminative feature representations can be learned. Finally, the proposed method is analyzed in the task of clustering on four datasets, and its effectiveness and superiority are validated through ablation studies, t-SNE visualization, and significance tests.

Although the mutual-information-based redundancy reduction constraints introduced in this study can effectively remove redundant information, they may also remove useful information to some extent. In future work, we plan to tackle this limitation by developing algorithms that can generate reliable pseudo-labels as ground-truth labels.

## Impact Statement

This work aims to advance self-supervised attributed graph clustering by improving graph topology reconstruction and node representation learning. The proposed method may benefit applications involving graph-structured data, such as citation network analysis, recommendation systems, social network mining, and biological network analysis. Since graph clustering methods can be applied to sensitive relational data, potential risks include privacy leakage, biased grouping, or unintended profiling when the input graph contains personal or socially sensitive information. These risks are not specific to our method but are common to graph representation learning and clustering techniques. We encourage practitioners to carefully examine data sources, privacy constraints, and fairness considerations before applying the proposed method in real-world scenarios.

## Acknowledgements

This work is supported by the National Natural Science Foundation of China (No.62402337) and the Postdoctoral Fellowship Program of CPSF under Grant (No. GZC20251059).

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
