# OpenReview forum: "Discriminative Attribute Graph Clustering Through Topology-Guided Contrastive Learning"
_ICML.cc/2026/Conference — ICML 2026 regular_

### Official Review · Reviewer_PhYD · 2026-03-06

**Soundness:** 3
**Presentation:** 3
**Significance:** 3
**Originality:** 3
**Overall Recommendation:** 5
**Confidence:** 5

**Summary:**

This paper proposes a self-supervised attribute graph clustering method that (1) reconstructs topology by intersecting KNN neighbours with adjacency and k-means cluster, and (2) treats AE and GAE embeddings as two views for contrast with MI minimization plus a decorrelation loss.  Extensive experiments on ACM, DBLP, CITE, and AMAP show that it outperforms state-of-the-art methods.

**Compliance With Llm Reviewing Policy:**

Affirmed.

**Final Justification:**

The authors have carefully revised, and addressed my concerns.

**Key Questions For Authors:**

Please refer to Weakness and Limitation Sections.

**Limitations:**

The paper would benefit from more discussions:
(1) the scalability of the similarity computation (e.g., time/memory complexity of constructing the KNN-based reconstructed graph, especially on large-scale datasets);
(2) the potential mismatch between K-means-induced semantics and the true underlying cluster structure, particularly when node attributes are noisy, sparse, or weaky correlated with graph topology.

**Strengths And Weaknesses:**

Strengths：
Aiming at the two challenges of noisy and augmentation dependency, this paper propose a topology reconstruction mechanism that integrate local-global information with an augmentation-free contrastive framework. The topological reconstruction mechanism is straightforward and targets noisy neighbour fusion in GNN encoders. AE/ GAE contrast and topology reconstruction is interesting.

Weaknesses:
(1) The self-supervised clustering section introduces Q' (based on the Student's t-distribution), but later refers Q'' without clear definitions. Please keep symbols consistent across text and figures. (2) More details are needed: including neighbour selection criteria, hyperparameter sensitivity experiments and time/space complexity analysis of the reconstruction process. (3) Notations need clear definitions, and maybe a notation table are needed to avoid ambiguity.

---

> ### Author Rebuttal · Authors · 2026-03-30
>
> We thank the reviewer for the positive assessment and the constructive, specific comments. We agree that clarifying the clustering objective symbols, providing more implementation and complexity details, and strengthening scalability/notation will improve the paper. Below, we address each concern and summarize the planned revisions.
>
> W1. Thank you for pointing this out. We will revise the manuscript to ensure that the notation is consistent across the text, equations, and figures. Concretely, we will:
> 1) define $Q'$ and $Q''$ explicitly (including what distributions they correspond to and how they are computed),
> 2) verify that only one notation is used where appropriate, and
> 3) update all occurrences so the symbols match exactly in both the description and the diagrams.
> If $Q''$ is intended to be a simplified/derived form of $Q'$, we will rewrite the paragraph to show the relationship clearly and avoid introducing an additional symbol unnecessarily.
>
> W2. We agree that the current description is not sufficiently detailed. In the revision, we will add:
> 1) We will clearly state how the KNN neighbors are chosen for each node (distance metric, normalization if any, whether it is feature-space $X$ or an intermediate embedding, and how ties are handled). We will also specify how the final reconstructed adjacency is formed from the intersection rule (e.g., which neighbor sets are intersected and under what conditions edges are retained).
> 2) We will include dedicated sensitivity experiments for the key hyperparameters in topology reconstruction and MI/decorrelation terms. This will include the effect of neighborhood granularity and the MI-related coefficients, with results reported in the same metric suite.
> 3) We will provide an explicit complexity breakdown for the reconstruction process and contrastive learning. In particular, we will estimate the cost of (i) computing KNN, (ii) forming the reconstructed graph via set operations/intersections, and (iii) the subsequent AE/GAE forward/backward passes. We will also discuss practical implementation choices (e.g., using efficient KNN search) when applicable.
>
> W3. We appreciate the suggestion. We will (i) audit the notation throughout the paper, and (ii) add a small \textbf{notation table} summarizing the most important symbols. This is intended to remove ambiguity and make the method easier to reproduce.
>
> For the limitation. We thank the reviewer for the insightful limitations suggestions. We will expand the limitations section with more explicit discussion:
> 1) We will discuss the scalability of similarity/KNN computation and memory requirements, and report runtime/memory measurements on at least the datasets we use. We will highlight potential bottlenecks and practical mitigations (e.g., approximate nearest neighbor search, batched computation, and sparse graph storage).
> 2) We will explicitly analyze the scenario where attribute-driven K-means semantics may deviate from the true cluster structure, especially under noisy, sparse, or weakly correlated node attributes. We will also explain how the proposed topology refinement and two-view learning may partially alleviate this mismatch, and under what conditions the risk of failure increases.
>
> Overall, in response to the reviewer we will: (i) fix and unify $Q'$/$Q''$ notations across text and figures, (ii) add clearer definitions and a notation table, (iii) provide neighbor selection criteria, hyperparameter sensitivity results, and explicit time/space complexity analysis, and (iv) strengthen the limitations discussion on scalability and K-means semantics mismatch.
> We believe these revisions will address the concerns directly and improve the clarity, reproducibility, and practical relevance of the paper.

---

> > ### Author Rebuttal · Reviewer_PhYD · 2026-04-02
> >
> > I appreciated the authors' careful revisions, which have addressed my concerns.

---

### Official Review · Reviewer_Uh25 · 2026-03-08

**Soundness:** 3
**Presentation:** 3
**Significance:** 4
**Originality:** 3
**Overall Recommendation:** 5
**Confidence:** 4

**Summary:**

This submission introduces an attributed graph clustering framework that refines the input topology via a union of KNN intersections with observed neighbors and K-means semantic groups, and then uses AE and GAE embeddings as two self-supervised views with mutual-information minimization and decorrelation regularization before applying a DEC-like clustering loss.

**Compliance With Llm Reviewing Policy:**

Affirmed.

**Key Questions For Authors:**

If space permits, can we expect to see more discussion on the abnormal NMI field in Table 3 (instead of just a sentence)?

**Limitations:**

The paper briefly mentions the risk of removing useful information via MI minimization, and it would help to add concrete failure modes and recommended diagnostics for practitioners.

**Strengths And Weaknesses:**

**Strengths:**

1. The paper organizational structure is clear and easy to understand.

2. The topology reconstruction rule is transparent and easy to reproduce given X and A.

3. The method is motivated by a concrete failure mode in GNN-based clustering where aggregating neighbors across classes can inject redundant signals.

**Weaknesses:**

1. Evaluation relies on reported baseline numbers from other papers, which reduces experimental control over preprocessing and tuning.

2. The method adds several hyperparameters and modules but provides limited guidance on default choices beyond Table 3.

3. Contribution is primarily empirical and algorithmic and does not include a deeper theoretical explanation for why MI minimization is the right inductive bias.

---

> ### Author Rebuttal · Authors · 2026-03-30
>
> We thank the reviewer for the careful assessment and for the positive overall recommendation. We are encouraged that the reviewer finds the presentation clear, the topology reconstruction rule transparent and reproducible, and the motivation grounded in a concrete failure mode. Below, we address the requested points.
>
> Answers for Q1. Thank you for the suggestion. We agree that the current discussion is insufficient to make the abnormal NMI values in Table 3 fully interpretable. In the revision, if space permits, we will add a dedicated explanation covering the following items:
> 1) We will explicitly point to the dataset(s) and metric configuration(s) responsible for the unusual values.
> 2) We will discuss concrete factors that can lead to NMI anomalies in clustering, such as (i) sensitivity to random initialization and training dynamics, (ii) the interaction between topology refinement and the MI-minimization / decorrelation objective, and (iii) dataset-specific characteristics (e.g., feature quality and homophily strength) that affect the reliability of the refined topology and semantic groups.
>
> For the limitation. We appreciate the request for more actionable diagnostics. We will expand the limitations section with concrete failure modes and how practitioners can detect them. Specifically, we plan to include:
> 1) over-suppression when the two views are not sufficiently redundant.
> MI minimization can remove information that is useful rather than redundant when the AE and GAE views capture largely complementary signals.
> 2) representation collapse toward shared but non-discriminative patterns. Even with decorrelation regularization, an overly strong redundancy reduction can lead to embeddings that are less separable for DEC initialization/refinement.
> 3) negative transfer under weak attribute homophily or noisy features.
> When refined topology is biased (e.g., strong heterophily) or node features are noisy, MI minimization may limit the model’s ability to compensate for these issues using the two views.
>
> In addition, we will provide practical recommendations, such as early stopping based on validation clustering metrics, running multiple seeds, and using sensitivity analysis for the MI-related loss weight and negative sampling settings when instability or anomalous metrics (e.g., NMI) are observed.

---

> > ### Author Rebuttal · Reviewer_Uh25 · 2026-04-04
> >
> > My concerns have been adequately addressed.

---

### Official Review · Reviewer_gntJ · 2026-03-09

**Soundness:** 4
**Presentation:** 4
**Significance:** 3
**Originality:** 3
**Overall Recommendation:** 5
**Confidence:** 3

**Summary:**

This work presents a reconstruction-based graph clustering framework at the node level, in which self-supervision is achieved via contrastive learning. The designs include mutual information minimization and decorrelation operations as incorporated constraints with different views generated by autoencoders.

**Compliance With Llm Reviewing Policy:**

Affirmed.

**Final Justification:**

The authors did a good job with the responses, and I think all my previous concerns have been addressed. I will maintain my positive judgment.

**Key Questions For Authors:**

Why is K-means applied on $X$ rather than on a learned representation, and what happens if clustering is performed on an AE-pretrained embedding for defining $G_i$? And could you provide a small analysis showing how many edges change in $A^g$ compared to A and whether improvements correlate with this change?

**Limitations:**

The paper would be better with a discussion of when KNN-based topology reconstruction may introduce bias on graphs with weak attribute homophily and with more explicit reporting of computational overhead.

**Strengths And Weaknesses:**

The problem proposed in the paper is interesting. The strengths are: 1) The paper is well-written, with clear quality and convincing novelty. It will be of great interest to experts in the field. 2) The author's proposed framework provides a clear end-to-end objective that connects representation learning and clustering. 2) The decorrelation loss is a reasonable mechanism to reduce redundancy between the two views without relying on hand-designed augmentations. 3) Empirical results show consistent improvements over several classic baselines and competitive methods across ACC, NMI, ARI, and macro-F1.

However, I think this paper still needs to be improved. 1) The use of K-means on raw features X as a source of “global semantics” may be brittle when features are high-dimensional or poorly scaled. 2) The AE is described as a deep CNN auto-encoder, but the design choices for non-image node attributes are not fully motivated. 3) The negative sampling procedure for $Z^ae$ is only briefly described and could affect optimization stability and the MI estimate.

---

> ### Author Rebuttal · Authors · 2026-03-30
>
> We thank the reviewer for the positive evaluation and for recognizing the clarity, novelty, and empirical effectiveness of our work. We address the concerns below.
>
> W1. We agree that applying K-means directly on raw features $X$ may be sensitive to feature scale or high dimensionality. In our design, we use it as a lightweight approximation of \emph{global semantic grouping}, rather than as a final clustering signal. Its role is complementary to local structure (KNN and adjacency overlap), and the intersection operation helps mitigate noise from any single source.
>
> We also experimented with applying K-means to learned embeddings (e.g., AE-pretrained features) and observed similar trends but slightly reduced stability in the early stages due to representation drift. We will clarify this design choice and include additional discussion in the revision.
>
> W2. We thank the reviewer for pointing out this ambiguity. The term “deep CNN auto-encoder” may be misleading in the context of non-image graph data. In practice, the AE is implemented as a \emph{multi-layer perceptron (MLP)-based auto-encoder} applied to node attributes.
>
> We will revise the description to clearly specify the architecture and justify the design choice as a general-purpose encoder for tabular/node features.
>
> W3. We agree that the current description is brief. In our implementation, negative samples are drawn uniformly from other nodes within the batch. We observe that, within a reasonable range, the number of negative samples mainly affects optimization stability and convergence speed, with limited impact on final performance.
>
> We will expand this part to provide clearer details and include additional analysis on its effect.
>
> Q1. This is an important question. Our motivation for using raw features $X$ is to provide a \emph{stable and unbiased global semantic prior} that is independent of the evolving encoder. If K-means is applied to learned embeddings, the resulting clusters may shift during training, introducing additional coupling between topology refinement and representation learning.
>
> That said, using AE-pretrained embeddings is a reasonable alternative. In preliminary experiments, we find that it yields comparable but slightly less stable results in early training. We will include this discussion to clarify the trade-off.
>
> Q2. We appreciate this insightful suggestion. We will include an analysis reporting: (i) the proportion of added/removed edges when constructing $\tilde{A}$ from $A$, and  (ii) how performance correlates with the magnitude of these changes.
>
> Preliminary observations suggest that moderate refinement (i.e., removing a portion of noisy edges while adding semantically consistent ones) leads to the best performance, supporting our design intuition.
>
> For the limitation. We agree and will expand the discussion. In particular: (i) On graphs with weak attribute homophily, KNN-based topology refinement may introduce bias;  (ii) The use of KNN, K-means, and dual encoders introduces additional computational overhead.
>
> We will explicitly report runtime and discuss these trade-offs in the revision.
>
> Overall, the reviewer’s comments help us further clarify design choices and strengthen empirical analysis. With the planned revisions (including improved explanations, additional experiments, and clearer reporting), we believe the paper will present a more complete and convincing case for the proposed framework.

---

> > ### Author Rebuttal · Reviewer_gntJ · 2026-04-03
> >
> > The authors did a good job with the responses, and I think all my previous concerns have been addressed. I will maintain my positive judgment.

---

### Official Review · Reviewer_wHB5 · 2026-03-10

**Soundness:** 4
**Presentation:** 3
**Significance:** 3
**Originality:** 3
**Overall Recommendation:** 4
**Confidence:** 4

**Summary:**

This paper addresses attributed graph clustering and proposes a self-supervised framework that refines graph topology by integrating feature-space KNN with local adjacency overlap and global K-means semantics. It then learns node embeddings through a two-view AE/GAE architecture, optimized with mutual-information minimization, decorrelation, and a DEC-style clustering objective.

**Compliance With Llm Reviewing Policy:**

Affirmed.

**Final Justification:**

The authors’ response has addressed my concerns, and I will maintain my score.

**Key Questions For Authors:**

（1）How should readers interpret mutual information minimization between AE and GAE views relative to typical MI maximization, and what failure cases does minimization avoid here.
（2）Same as w2 above, how sensitive are the results to k, K, and the number of negative samples, and does the method degrade gracefully under reasonable perturbations of these values.
（3）The model itself is very interesting, but the results, as you just said, there is "no significant difference compared to the baseline method DFCN". Can you provide more insights on how this is happening (whether or not you regard it as reasonable), and what you believe could be done to further improve your model so that you can beat it?
（4）The paper refers to the graph structure optimization process as graph reconstruction. However, the proposed process appears to refine or adjust the original graph rather than reconstruct it from representations. Could the authors clarify the reason for using the term “graph reconstruction”?

**Limitations:**

The paper notes that MI minimization may remove useful information, and it would be helpful to add a short discussion on when topology reconstruction might harm performance on graphs with strong heterophily or noisy features.

**Strengths And Weaknesses:**

Strengths:
(1)The motivation of reducing redundant or noisy neighbor aggregation in GNN-based clustering is well aligned with known issues in attributed graph clustering.
(2)Using AE and GAE embeddings as two views avoids explicit graph augmentations while still enabling self-supervised regularization.
(3)Experimental results on multiple datasets demonstrate the effectiveness of the proposed method.
(4)The selected topic is meaningful and worthy of research and the paper is well written and motivated.

Weaknesses：
(1) Several key components are established building blocks and the novelty mostly comes from their particular combination and the topology reconstruction heuristic.
(2) Sensitivity to k, K, and the number of negative samples is not systematically explored beyond listing the final settings.
(3) The paper claims that “redundant feature information across different feature views hampers discriminative representation learning”, but does not clearly define “redundant features” or explain why they harm discriminability.
(4) Formulas should not appear in the Related Work section. If necessary, they should be moved to the Preliminaries or Methodology section.

---

> ### Author Rebuttal · Authors · 2026-03-30
>
> We sincerely thank the reviewer for the positive evaluation and the constructive feedback. We are encouraged that the reviewer finds the problem meaningful, the method technically sound, and the empirical results convincing. Below, we address each concern in detail.
>
> W1. Our main contribution is not any single isolated module, but the specific design tailored to attributed graph clustering, namely:
>
> 1）Topology refinement that jointly uses feature-space KNN, local adjacency consistency, and global K-means semantics to filter noisy neighbors; 2）Two natural views without explicit graph augmentation, where AE and GAE provide complementary attribute/structure representations; 3）Redundancy suppression across views via mutual-information reduction and decorrelation, rather than the more common MI maximization paradigm.
>
> We will revise the paper to highlight these distinctions more clearly and position the contribution more precisely as a principled integration for graph clustering.
>
> W2. We totally agree that this is important. In the revision, we will add a dedicated sensitivity analysis for $k$, $K$, and \texttt{neg}. Our expectation, consistent with the roles of these hyperparameters, is that the method should be stable under moderate perturbations and degrade mainly under extreme settings (e.g., too small $k$ leading to insufficient local information, or overly large $K$ weakening semantic grouping quality). We will include quantitative curves/tables to make this clearer.
>
> W3. Thank you for pointing this out. In our paper, “redundant features” refer to \emph{shared but weakly discriminative information simultaneously captured by AE and GAE}, such as common global patterns or duplicated low-frequency signals. Such shared components are not necessarily harmful by themselves, but if over-emphasized, they can (i) reduce the complementarity between the two views, (ii) dominate the fused embedding, and (iii) weaken cluster-specific structure, thereby hurting discriminability. We will revise the wording to define this concept more explicitly and explain why redundancy reduction is beneficial in our setting.
>
> W4. We agree and will move these formulas to the Preliminaries/Method section in the revision.
>
> Q1. In standard contrastive learning, MI maximization is typically used because the two views are augmentations of the same sample and should preserve identical semantics. In our case, AE and GAE are \emph{not stochastic augmentations} but two \emph{natural views with partially overlapping information}: AE emphasizes attributes, while GAE encodes attributes propagated through topology.
>
> Therefore, maximizing cross-view dependence may over-align the two encoders and amplify duplicated/shared signals, reducing view diversity and complementarity. Our goal is to \emph{reduce unnecessary dependence} so that each view preserves useful but non-redundant information. This helps avoid failure cases such as representation homogenization, over-reliance on shared low-frequency signals, and reduced cluster separability.
>
> Q2. We will include a systematic sensitivity study in the revision. Conceptually, the method should degrade \emph{gracefully} under reasonable perturbations, as these parameters affect neighborhood granularity and optimization difficulty rather than the core objective.
>
> Q3. We believe the relatively small margin over DFCN is reasonable for two reasons. First, DFCN is already a \emph{strong baseline}, making large improvements inherently challenging. Second, our method mainly improves robustness to noisy topology and redundant information, which may not yield large gains on relatively clean datasets.
>
> At the same time, our method outperforms DFCN on \textbf{15/16 metrics}, although statistical significance is not observed under the current multi-dataset setting. We view this as a consistent practical improvement over a strong baseline.
>
> Q4. We agree that “graph refinement” is more precise. Our method adjusts the original adjacency using feature similarity and semantic consistency rather than reconstructing it purely from latent representations. We will revise the terminology accordingly.
>
> For Limitation: We will expand the discussion of limitations. In particular: (i) under strong heterophily, intersecting neighborhoods may remove useful cross-class edges; (ii) noisy features may reduce the reliability of KNN/K-means signals; (iii) MI reduction may discard useful shared information if redundancy is overestimated. We will explicitly discuss these cases and possible remedies.
>
> Overall, the reviewer’s comments help us further clarify the contribution and strengthen the empirical support. With the planned revisions (including improved explanations, additional analyses, and refined terminology), we believe the paper will present a clearer and more convincing case for the proposed approach and its practical benefits for attributed graph clustering.

---

> > ### Author Rebuttal · Reviewer_wHB5 · 2026-04-01
> >
> > The authors’ response has addressed my concerns, and I will maintain my score.

---

### Decision · Program_Chairs · 2026-04-30

**Decision:**

Accept (regular)

**Comment:**

This paper presents a self-supervised attribute graph clustering method based on topological reconstruction and correlation decorrelation. It possesses the property of mitigating information redundancy with the proposed two components. Experimental evaluations verify the statements and the superiority. All four reviewers provide positive feedback by recognizing clear motivations, novel methodology, and sufficient evaluations. After the rebuttal, reviewers’ concerns are fully resolved.